# Resource and Environmental Pressures on the Transformation of Planting Industry in Arid Oasis

**DOI:** 10.3390/ijerph19105977

**Published:** 2022-05-14

**Authors:** Jing Huang, Dongqian Xue, Chuansheng Wang, Jiehu Chen

**Affiliations:** 1School of Geography and Tourism, Shaanxi Normal University, Xi’an 710119, China; huangjing121908@163.com; 2Institute of Geographic Sciences and Natural Resources Research, Chinese Academy of Sciences, Beijing 100101, China; wangcs@igsnrr.ac.cn; 3School of Computer Science, Shaanxi Normal University, Xi’an 710119, China; jiehu90526@163.com

**Keywords:** coupled linkage relationships, environmental pollution load, Ganzhou district, planting industry development, water resource demand

## Abstract

Controlling environmental pollutant discharge and water resource demand is crucial for the sustainable development of agriculture and rural areas in arid oases. Taking Ganzhou, an arid oasis in Northwest China, as an example, we established an analysis framework for the relationship between the planting industry transformation and the resource and environmental pressures, from 2011 to 2020, through the methods of inventory, coefficient and quota accounting. The results showed that the planting scale of crops in oases has continuously expanded, with a structural dominance of corn seed production. Pollutant discharge showed a “Z”-type evolution trend, and the demand for water consumption continued to increase. The transformation of the planting industry and pollutant discharge showed coupled trade-offs and a synergetic alternating fluctuations coupling relationship, which was highly co-evolutionary with the demand for water resources. Crop planting exhibited four spatial patterns, namely the mixed planting area of grain and cash crops grown in mountain areas (GCPA), suburban scale vegetable planting (SVPA), planting of seed production corn (MSPA), and the compound planting area of grain crops, oil crops, vegetables, and other characteristic crops (CMPA). MSPA and SVPA had the highest total and average volume per unit area, respectively. The planting industry transformation and evolution of resource and environment pressures are closely related to changes in national strategies, regional agricultural policies, and environmental regulations. Therefore, studying their relationships provides a scientific basis for the formulation of suitable countermeasures, according to the development stage of a region.

## 1. Introduction

Arid areas account for 41% of the global land area and support approximately 38% of the population. The economy, society, and sustainability of these regions face many challenges, since water-deficient ecosystems are fragile and sensitive to disturbance by extreme climate and anthropological activities [1]. An oasis is a region in arid areas where there is water. For example, the Heihe River, the second largest inland river in Northwest China, originates from the northern foot of the Qilian Mountain and is replenished by glacier snowmelt. It nurtures the Zhangye oasis in the middle reaches of the Hexi Corridor, which supports 95% of the cultivated land, 91% of the population, 83% of the water consumption, and 89% of the GDP in this river basin. It is an important production base for grain, oil, and vegetable crops in China, and is one of the arid areas with the best resources, densest population, and richest agriculture. Further, agricultural modernization has excessively relied on the input of chemicals [2] because of the continuously expanding scale of cultivated land resource development in the middle reaches over recent years [3]. Although the agricultural economy has developed rapidly, unreasonable land reclamation, water use structures, and extensive irrigation have led to a pronounced discrepancy between supply and demand for insufficient water resources. This leads to the overexploitation of the groundwater in the river basin, which currently allows for the normal functioning of the system. This is coupled with production and domestic water use, and the pollution of the water and soil environment by agricultural production. Therefore, the oasis ecological environment, which is linked by water resource relationships, has begun to deteriorate and endanger regional ecological security [4].

The development of rural areas in China has entered a new stage of comprehensive revitalization and transformation to move toward modernization. The coordinated high-quality development of the social economy, resources, and the environment in rural areas with green development is the current trend [5]. In agriculture, the focus has shifted from increasing production to improving quality, which is essential for improving the rural environment [6]. However, water resource use at the expense of the ecological environment in the oases of arid lands in China is becoming increasingly prominent [7], because it also hinders the sustainable development of agriculture and rural areas. Therefore, an urgent scientific assessment of agricultural development and its resource and environmental pressures is required.

This study focuses on Ganzhou district, an oasis-irrigated agricultural area, to explore the relationship regarding the transformation of the planting industry with resource and environmental pressures. The most significant contributions of this study are listed below: Firstly, an evaluation framework for the relationship between the planting industry transformation and resource and environment pressures in arid oases was established. Secondly, inventory analysis and coefficient methods were used to calculate the environmental pollution load and water resource demand at the county and township scales. Lastly, changes in resource and environmental pressures caused by the transformation of the planting industry were also investigated using the link analysis method. Importantly, machine learning was used to divide the functional areas of the crop planting patterns, and the differences in resource and environment pressures caused by different planting patterns were studied. The research results could provide important insights and a basis for the sustainable development of arid oasis agriculture and rural areas.

## 2. Research Review and Framework

### 2.1. Literature Review

While studying the relationship between rural area development with resources and the environment, researchers worldwide have predominantly focused on the following aspects: (1) the impact of agricultural production and rural economic development on resources and the environment; (2) impact of urbanization and industrialization on rural resources and the environment; (3) measurement and evaluation of resource and environment impact indices, and (4) protection mechanisms.

#### 2.1.1. Impact of Agricultural Production/Rural Economic Development on Resources and Environment

Agriculture is a complex socio-economic natural ecosystem [8]. The impact on resources and the environment mainly originates from the planting industry, breeding industry, and residents’ living. This impact can be measured by the input of chemical fertilizers, pesticides, and agricultural films, emissions of carbon dioxide (CO_2_), chemical oxygen demand (COD), ammonia nitrogen (NH3-N), total nitrogen (TN), and total phosphorus (TP) [9,10]. Wood biochar has shown excellent effects for increasing soil and plant-available water, nutrients and energy management, especially in transforming phosphorus and nitrogen into bioavailable forms [11]. In most rural areas, agricultural production and economic development are based on highly intensive chemical input [12], which has a major impact on resources and the environment. This leads to problems, such as soil fertility damage, soil hardening, and groundwater pollution [13]. Empirical research on the relationship between agriculture and economic growth and environmental quality by the United States and the European Union has shown this relationship is complex [14]. Heavy resource and environmental costs were paid owing to the extensive use of chemical fertilizers and pesticides in early agricultural modernization in Japan [15]. The high investment coupled with the low utilization and recovery rate of chemical fertilizers, pesticides, and mulching films in intensive and large-scale agricultural production in China has resulted in large volumes of surplus agricultural chemicals being lost through runoff. This has led to severe water and soil pollution in many areas [16]. For example, the nitrate pollution of groundwater has occurred in Shandong Province, Henan Province, and other major grain-producing areas in East China [17]. Furthermore, the average rate of chemical fertilizers in Gansu province in Northwest China was approximately 30% [18] and the residual amount of mulching film per unit area of cultivated land in Xinjiang was four to five-times the national average [19]. The evolutionary trend of pollutants during different stages of rural economic development has been heterogeneous [20]. In underdeveloped countries, the level of economic modernization has been directly related to the input of agricultural chemicals [21]. However, in both developed and developing countries, research on the relationship between agriculture and the environment has predominantly focused on carbon dioxide emissions rather than non-point source pollution emissions [22]. The relationship between agricultural production and rural economic development with resources and the environment is predominantly based on the empirical test of the theoretical hypothesis of the environmental Kuznets curve (EKC). Most studies demonstrated that they follow an inverted U-shaped coupling trajectory [23], whereas some studies have shown coupling paths, such as an N-shape or inverted N-shape [24,25]. Some researchers have also highlighted that the influences between these paths are multidirectional, intertwined, and dynamic. These may form three types of coupling relationships, namely rising, inverted “U,” and falling [26].

#### 2.1.2. Impact of the Urbanization/Industrialization Process on Rural Resources and the Environment

Due to the influence of Keynesian economic development determinism, as early as the 1950–60s, Western countries ignored the environmental damage that accompanied urbanization and industrialization. This led to the emergence of “the world’s eight pollution nuisances.” Following this, many quantitative studies explored the relationship between development and the environment [27]. Although the early stages of urbanization usually lead to the deterioration of the agricultural environment, practices in most Western countries have shown that high levels of urbanization are the only way to achieve an optimal environment [28]. China’s urbanization started relatively late. However, the rapid process has led to benefits that have attracted attention worldwide. However, this developmental path, highly dependent on land resources, has not only led to the wastage and destruction of resources but also to severe ecological and environmental problems [29,30]. Poorly planned expansion of construction land has encroached on rural cultivated land, forests, and water bodies. This led to the flow of rural labor and other resources to cities, destruction of the environment, weakening of regional functions, and aggravation of system vulnerability [31,32]. The resource and environmental effects of industrialization in the urban–rural transformation process in China have had a profound impact on the production and residential activities in rural areas. Industrialization with township enterprises has driven economic development. However, it has resulted in resource waste, environmental pollution, and ecological damage due to scattered layouts, small scales, and industrial isomorphism [33]. Therefore, researchers have conducted many studies on the coordination relationship between urbanization/industrialization with the resources and environment [34,35,36]. Many researchers highlighted that economy and society are closely related to the coordination among resources, the environment, and sustainable development, with certain patterns needing to be followed during regional development to achieve synchronous development [37,38,39]. Due to differences in the level of economic development of different regions, the influences of urbanization/industrialization on resources and the environment are varied; East China presented an inverted “U” curve relationship, whereas the Central and Western regions showed contrasting results [40]. Therefore, high-quality coordinated development can be achieved only by formulating targeted strategies.

#### 2.1.3. Measurement and Evaluation of Resource and Environmental Effect Indices and Protection Mechanisms

Research on the quantitative evaluation of resource and environmental effects has continuously improved with developing GIS, RS, computer simulation, and other technologies. This predominantly includes index systems or comprehensive indices, measurement models, GIS analysis, and simulation methods [41,42]. The corresponding index or index system has mainly been used when evaluating the impact of a single factor on resources and the environment [43,44,45]. Econometric models were often applied for quantitative research to systematically evaluate and simulate the effects of various factors on resources and the environment [46,47,48]. In developed countries, rural governance and resource and environmental protection policies often have seamless legal systems to control development in rural areas and to protect the environment. In contrast, China’s land and spatial planning mainly focus on rural governance strategies [49]. Many countries have used fiscal policies to prevent and control environmental problems and cultivate awareness among citizens of environmental protection through environmental education. They have also built a rural environmental culture dominated by sustainable development, reducing unnecessary agricultural pollution, ensuring the safety of agricultural products, promoting ecological agriculture, using energy-saving and environment-friendly materials, and promoting garbage sorting, among other measures, to improve rural living conditions [50,51,52]. This has shown that spatial governance by multiple means, such as regulation, planning, trading of pollution discharge rights, and autonomy of residents, is conducive to protecting resources and the environment in rural areas [53,54].

### 2.2. Research Framework

The conflict between environment and development has been prominent throughout the process of global rural area transformation and urbanization/industrialization. Furthermore, people have been paying increasing attention to environmental pollution. Therefore, the empirical analysis of the effect of pollution emissions on the relationship between rural area development and the environment has been comprehensively reflected in current research. The radical urbanization of China at the expense of rural areas has caused severe resource and environmental challenges. This has also promoted research on resource and environment effect evaluations, which focus on land resources and effects on the ecological environment. Although the relationship between rural area development and the resources and environment has been extensively explored, there are still multiple areas for future research. Many prior studies have had data-availability limitations and investigated the pattern characteristics and evolutionary path of the relationship between rural area development with resources and environment at a specific scale, at or above the county level. The village scale can better reflect the characteristics of rural areas but has remained relatively unexplored. There has also been a lack of comparison and consideration of scales. Most studies established an index system and adopted the synthetic index method for evaluations, whereas the clear correspondence and evolution logic were rather vague. Therefore, it is necessary to establish the relationship between element structure changes of resources and the environment from a more detailed perspective. Most studies focused on developed areas with rapid urbanization in East China, or areas with a severe decline in the rural environment in Central and West China, while ignoring the arid and densely populated oasis agricultural areas in Northwest China.

The processes of urbanization, rural revitalization, and agricultural modernization are leading to fundamental changes in the relationship among human activities, resources, and the environment; they are intensifying the changes in the human–land relationship patterns in rural areas of China [55,56]. The relationship of rural area development with resources and the environment in oasis agricultural areas is characterized by the ecological fragility and high-intensity production in arid areas. This may highlight different characteristics and evolutionary rules compared with other regions in China. However, the contradiction of agricultural development with water resources and the ecological environment is pronounced, owing to insufficient long-term focus. Furthermore, clarifying the interactions of development with resources and the environment is important for the sustainable development of agricultural areas in oases of arid lands in theoretical and practical needs. In these areas, any production and residential activities are restricted by limited water resources. The unplanned expansion of cultivated land and unsustainable water use infrastructure are the key causes of resource and environmental problems. It is necessary to focus on transformation of the planting industry and the relationship between water resources and environmental pressure to improve ecological security in these areas. Based on a combined relationship between planting industry transformation, pressure on the environment, and water resource demand, this paper develops an analysis framework that integrates scale, structure, and spatial patterns at the county and township scales (Figure 1). This study also investigates pressures associated with water resources and the environment caused by planting industry transformation in oasis agricultural areas. This research aims to clarify the linear or non-linear relationships between rural area transformation with resources and the environment, and provides policy suggestions for the sustainable development of agriculture and rural areas in these regions.

## 3. Materials and Methods

### 3.1. Overview of the Study Area

Ganzhou district is located in the arid region of Northwest China (38°32′–39°24′ N, 100°06′–100°52′ E). It belongs to the Zhangye City, Gansu Province, and is located in the middle of the Hexi Corridor. The altitude of the district is 1370–3637 m. It has the Qilian Mountain to the south, Heli and Longshou Mountains to the north, and a corridor plain in the center. The terrain is high in the south and low in the center and it has a unique corridor topography and desert oasis landscape. The entire district is 65 km long from east to west, 98 km wide from north to south, and has a total area of 3661 km^2^. It has authority over five subdistricts and 18 townships (Figure 2). Ganzhou district was chosen as the case area because this area has relatively low levels of precipitation, an arid climate, and a fragile ecological environment. Water resources in the area are predominantly derived from ice and snowmelt in the southern Qilian Mountains, with regional development being substantially influenced by water resource constraints. It is also an important oasis agricultural area in China. In 2020, the rural population represented 46.43% of the total population, with high population density in agricultural areas, and well-developed irrigation farming. It is the largest county-level corn seed production base in China, a major agricultural hub at the national level, and a key development zone in the Gansu province dominated by agriculture. Over the last decade, the cultivated land area has expanded by 39.01%, with the agricultural water consumption remaining at approximately 90% of the total water consumption. Excess agricultural water consumption has eclipsed other forms of water consumption, resulting in groundwater overexploitation, endangering regional water and food supply and ecological security. Agricultural production depends on chemical use, including fertilizers, pesticides, and mulching films. This causes serious non-point source pollution, which poses a severe threat to the regional ecological environment. In conclusion, Ganzhou district encompasses the most common problems associated with the sustainable development of agricultural areas in oases of arid lands in China. It is, thus, an ideal case for studying the transformation of the planting industry and its resource and environmental pressures. The dynamic laws of the development of agricultural and rural areas in Ganzhou district can be extrapolated to other similar areas.

### 3.2. Research Methods

#### 3.2.1. Crop Planting Dominance Index

The dominance index refers to the degree of specialization of a certain crop in a township [57]. It can be used to analyze the comparative dominance of different crops in a township, or that of the same crop in different townships. The specific formula for calculating crop planting dominance index is as follows:(1)$$PCAI_{mi}=\frac{CSA_{mi}/CSA_{m}}{CSA_{i}/CSA}$$
where *PCAI_mi_* represents the comparative dominance index of crop *i* in township *m*, *CSA_mi_* is the planting area of crop *i* in township *m*, *CSA_m_* is the total planting area of crops in township *m*, *CSA_i_* is the total planting area of crop *i* in the district, and *CSA* is the total planting area of crops in the district. *PCAI_mi_* > 1 suggests that the production scale of crop *i* in township m is higher than the level in the district, whereas *PCAI_mi_* < 1 suggests that the production scale of the crop *i* in township m is lower than the level in the district. Therefore, a higher *PCAI_mi_* value indicates a higher level of dominance at the production scale.

#### 3.2.2. Calculation of Environmental Pollution Load

The discharge loss of pollutants (*NH*_3_-*N*, *TN*, *TP*) from the planting industry was calculated using the production and discharge coefficient method [58]. The formulas for calculating the discharge loss of various pollutants are as follows:(2)$$Q_{NH_{3}-N}=\left(A_{g}\times e_{gNH_{3}-N}+A_{y}\times e_{yNH_{3}-N}\right)\times \frac{q_{NH_{3}-N}}{q_{0}}$$
(3)$$Q_{TN}=\left(A_{g}\times e_{gTN}+A_{y}\times e_{yTN}\right)\times \frac{q_{TN}}{q_{0}}$$
(4)$$Q_{TP}=\left(A_{g}\times e_{gTP}+A_{y}\times e_{yTP}\right)\times \frac{q_{TP}}{q_{0}}$$
where *Q_NH3-N_*, *Q_TN_*, and *Q_TP_* are the discharge losses of pollutants *NH_3_-N*, *TN*, and *TP* from the planting industry, respectively. *Ag* refers to the total planting area of crops, *eg_NH3-N_*, *eg_TN_*, and *eg_TP_* are the loss coefficients of pollutants *NH_3_-N*, *TN*, and *TP*, respectively, during crop planting. *Ay* refers to the garden plot area; *ey_NH3-N_*, *ey_TN_*, and *ey_TP_* are the loss coefficients of pollutants *NH_3_-N*, *TN*, and *TP*, respectively, in the garden plot. *q_NH3-N_*, *q_TN_*, and *q_TP_* represent the amounts of nitrogen and phosphorus fertilizers. These are used in the planting industry per unit area in the research year after conversion to the number of active ingredients. *q*_0_ represents the usage amount per unit area of nitrogen and phosphorus-containing fertilizers used in the planting industry in 2017 after conversion of the number of active ingredients (Table 1).

#### 3.2.3. Water Resource Demand Calculation

The water consumption of sustainable agricultural production encompasses two components on the premise of ensuring normal production in the planting industry with a clean environment. These are: (1) water for crop growth and (2) water for pollutant purification [59]. Water for crop growth is expressed as the product of crop planting area and irrigation norm, whereas water for pollutant purification is expressed as the maximum water demand of water environment carrying capacity. The calculation formula for the water demand for crop growth is given below:(5)$$APWC_{m}=\overset{n}{\underset{i=1}{\sum }}A_{mi}\times a_{i}$$
where APWC_m_ refers to the water demand for crop growth in township m, n is the crop species, A_mi_ is the planting area of crop i in township m, and a_i_ is the irrigation norm for crop i (Table 2).

The pressure of planting in water environments is the discharge of excess nutrients, such as nitrogen and phosphorus. In this study, the maximum excess nitrogen load of the surplus water is an index to characterize the carrying capacity of water environments. The specific formula for water demand for pollution purification during planting is as follows:(6)$$PPWC_{m}=N_{m}/\left(N_{max}\right)$$
where PPWC_m_ is the water demand for pollution purification in the planting industry in township m, N_m_ is the nitrogen discharge loss of planting industry in township m, which is expressed as TN, and N_max_ is the standard for the water environment carrying capacity. The recommended standard in EU agricultural policies is 50 mg/L. In addition, 0.4 is the critical value for the water environmental pollutant load alarm; when the value exceeds 0.4, the nitrogen content in the water body will threaten the environment.

#### 3.2.4. Response Intensity Model

A resource and environment response intensity model was established to clarify the coupled linkage relationship of the planting industry with resource and environmental pressures. The response pattern of environmental pollution discharge and changes in water resource demand during the planting industry transformation was analyzed. The model is as follows:(7)$$RS=\left[\left(\Delta RE/RE_{i}\right)/\left(\Delta PI/PI_{i}\right)\right]$$
where RS represents the response intensity of resources and environment during the planting industry transformation. RE_i_ represents the environmental pollution discharge and water resource demand during period 6 > i, and ∆RE = RE_T__+I_ − RE_i_ represents the change in environmental pollution discharge and water demand during period T. PI_i_ represents the planting area of crops during period i and ∆PI = PI_T__+I_ − PI_i_ represents the change in the planting area of crops during period T. According to the changes in the planting industry and the response intensity of resources and environment, they can be divided into the following coupled linkage patterns (Table 3).

#### 3.2.5. K-Means Algorithm

The K-Means algorithm is an unsupervised clustering algorithm based on machine learning, which has been widely applied because of its simple implementation and satisfactory clustering effect [60]. This study used the K-Means algorithm to divide the functional areas of planting patterns. The data expression is as follows:

Given a sample set $D=\left\{x_{1},x_{2},...x_{m}\right\}$, the cluster obtained by clustering is divided into $C=\left\{C_{1},C_{2},...C_{k}\right\}$, and the goal is to minimize the square error *E*:(8)$$E=\overset{k}{\underset{i=1}{\sum }}\underset{x\in C_{i}}{\sum }x-\mu_{i}{}_{2}^{2}$$
where *μ_i_* is the mean vector of the cluster *C_i_*, and the expression is as follows:(9)$$\mu_{i}=\frac{1}{\left|C_{i}\right|}\underset{x\in C_{i}}{\sum }x$$
where *E* represents the closeness of the samples around the cluster mean vector. A smaller value of *E* indicates higher similarity between the samples in the cluster. To find the minimum value of *E*, all possible cluster divisions of the sample set *D* should be examined. Therefore, the K-Means algorithm uses an iterative heuristic approach to obtain the optimal value.

### 3.3. Data Sources

The planting areas of crops in the townships were derived from Ganzhou Agricultural Statistics Annual Report (2011–2020). Data on the amounts of chemical fertilizers, pesticides, and mulching film used in Ganzhou district were based on the Ganzhou Statistical Yearbook (2011–2020). The nitrogen and phosphorus discharge loss coefficients for the planting industry were established based on the Handbook of Calculation Methods and Coefficients of Pollutant Production and Discharge in Emission Source Statistical Investigation (https://www.mee.gov.cn/xxgk2018/xxgk/xxgk01/202106/t20210618_839512.html, accessed on 11 June 2021) issued by the Ministry of Ecology and Environment of the People’s Republic of China. The irrigation quota for crop growth in the Hexi area was determined from the agricultural water quota in the local standard Industry Water Quota (DB62/T 2987) of Gansu province issued on 7 March 2019, and was revised by combining these data with results from field investigations in the townships.

## 4. Results

### 4.1. Characteristics of Planting Industry Transformation

By analyzing the planting areas and proportions of crops in Ganzhou district from 2011 to 2020 (Figure 3), the planting scale of crops continued to expand from 56,967 to 73,280 hm^2^, with the proportion of grain crops remaining above 75% for many years. Therefore, it was a grain-based planting area. Corn was the most dominant food crop, with the proportion of its planting area continuously increasing from 60.75 to 82.56%. The planting area for wheat and potatoes continued to decrease, with the proportion of the planting areas reducing from 9 to 2.56% and 4.77 to 0.26%, respectively. Beans and other grains accounted for a low proportion and were characterized by intermittent fluctuations. Vegetables were the dominant cash crop, and their planting area remained between 100,000 and 200,000 mu for many years. Oil crops, sugar beets, fruits, and flowers accounted for a relatively small and stable proportion, whereas green feed and other crops were greatly reduced. In recent years, the planting area and the proportion of medicinal materials has increased continuously from 11 to 1244 hm^2^, with the proportion increasing by nearly 100-times.

From the perspective of spatial distribution and change characteristics of crop planting dominance (Figure 4), townships with grains as the dominant crops expanded from the south and north to the east and west over a large area. The western region had prominent advantages in corn seed production. This was followed by wheat and other grains, which were mainly distributed in townships along the southern mountain. The spatial extent of potatoes, oil crops, sugar beets, and other crops contracted across a large area, especially in the southeast. In recent years, the planting scope of beans has substantially expanded, and marginal towns have been planting oil crops. Vegetable planting was widely distributed, mainly on the city edge. Some townships were characterized by fruits and flowers planted over relatively small areas. Owing to the introduction of medicinal crops in southern towns, these crops expanded to other townships on a large scale. Green feed was planted mainly in the central and northern townships, with its proportion decreasing over time. The extent of other crops shrunk from south to north until they disappeared completely. In general, the grain crops in Ganzhou district were widely distributed and the planting scale was much larger than that of cash crops. Overall, the spatial distribution showed a dominance distribution pattern of grains in the northwest, vegetables on the edge of the city and the east, and cash crops in the southern and marginal towns.

### 4.2. Resource and Environmental Pressures

#### 4.2.1. Environmental Pressure

Crop production in Ganzhou district was overly dependent on the input of agrochemicals. Figure 5 shows the requirements of chemical fertilizers, pesticides, and mulching films per unit area of cultivated land from 2011 to 2020. Although there has been a decreasing trend over the last 10 years, the consumption was still high, and the utilization rate was low. Average fertilizer consumption exceeded the internationally recognized upper safety limit (225 kg/hm^2^) by 1.58–2.5-times, whereas the average utilization rate was only approximately 30%. The average consumption of pesticides had been stable at 3–5 kg/hm^2^ for many years, which is approximately 1.5–2-times that of the whole of Zhangye City. The average consumption of plastic film per unit area of land fluctuated between 30 and 60 kg/hm^2^, whereas the recycling rate was only 70%.

Figure 6 shows the changes in pollutant discharge loss from crop planting in Ganzhou district from 2011 to 2020. NH3-N, TP, and TN initially increased slightly, then decreased sharply after reaching a peak in 2015, before becoming more stable. During the study period, the discharge of pollutants showed a “Z”-shape evolution trend. NH3-N increased from 3277.56 to 3806.06 kg, then decreased to 2050.58 kg, before becoming stable at over 2000 kg. TP increased from 2300.16 to 2964.50 kg and then decreased to 1516.71 kg. The TP then increased slightly and stabilized at over 2100 kg, displaying the smallest range of change. TN pollution contributed the most, with discharge loss remaining at a high level. It increased from 44,792.22 kg in 2011 to 52,099.92 kg in 2014, and then substantially decreased before stabilizing at around 28,000 kg. In the study period, 2016 was an important turning point for planting pollution, mainly related to the notice “Action Plan for Zero Growth of Fertilizer and Pesticide Use by 2020”, issued by the Ministry of Agriculture, China, in 2015. In response to this call from the state, Ganzhou district follows a green agricultural development route by reducing fertilizers, conserving resources, and improving environmental sustainability.

#### 4.2.2. Water Resource Pressure

Ganzhou district has a high demand for water consumption for crop production, and Figure 7 shows changes in demand for water resources for crop planting from 2011 to 2020. Water demand for crop growth showed a continuous growth trend with the expansion of cultivated land and the planting area of crops, from 363.63 to 444.944 million m^3^, representing an increase of 22.36%. With changes in pollution discharge loss during crop planting, the demand for water for pollution purification also presented a “Z” evolution feature. It was stable between 2 and 2.5 million m^3^ before 2015 and remained at over 1.4 million m^3^ from 2016. The total water demand for the planting industry showed a steady growth trend, with an average annual growth rate of 2.29% and a total increase of 80,471,700 m^3^ over 10 years.

### 4.3. Coupling Relationship between Planting Industry Transformation with Resource and Environmental Pressures

#### 4.3.1. Coupling Characteristics of Planting Industry Transformation with Resource and Environmental Pressures

According to the relationship of planting industry transformation with pressure responses for resources and the environment (Table 4), some observed that pollution discharge showed response characteristics of alternating fluctuations between trade-off and synergy, with changes in the planting scale and structure of crops in Ganzhou district. Meanwhile, the demand for water resources commonly showed a positive synergetic response. The discharge of environment pollution decreased with planting industry transformation from 2011 to 2020, whereas the demand for water resources increased.

#### 4.3.2. Differences in Resource and Environment Pressure under Different Crop Planting Patterns

To further examine resource and environmental pressure differences under different spatial planting patterns, the K-Means clustering algorithm was used to divide functional areas according to the scale of planting and the proportion of the area used for various crops in townships, from 2011 to 2020. According to the clustering results of the K-Means algorithm (Figure 8), the planting patterns of crops in townships can be divided into four types, namely the mixed planting area of grain and cash crops grown in mountain areas (GCPA), suburban scale vegetable planting (SVPA), planting of seed production corn (MSPA), and the compound planting area of grain crops, oil crops, vegetables, and other characteristic crops (CMPA). The characteristics of each type are given in Table 5.

The environmental pollutant discharge load and water demand under different crop planting patterns were significantly different (Figure 8). In 2011–2020, the MSPA pattern contributed the most to total pollutant discharge, and was, on average, 6.81, 3.30, and 1.63-times that of the GCPA, SVPA, and CMPA patterns, respectively. The SVPA model had the highest average pollutant discharge, which was, on average, 1.49, 1.15, and 1.18-times that under the GCPA, MSPA, and CMPA patterns, respectively. The total amount of water demand under MSPA and the amount per unit of land under SVPA were still the highest. The total water demand for the MSPA pattern was, on average, 6.87, 3.59, and 1.64-times that of the GCPA, SVPA, and CMPA patterns. The average demand for water resources under the SVPA pattern was 1.36-times that of the lowest GCPA pattern, whereas those from the MSPA and CMPA patterns were equal. Pollutant discharge for different planting patterns showed two evolutionary trends, namely an inverted U-shape and continuous decline. Meanwhile, there were three evolutionary trends for water demand, namely an inverted “U” shape, continuous decline, and continuous increase.

## 5. Discussion

### 5.1. Oasis Rural Area Development and Resources and Environment

Resources and the environment in rural areas have received increasing attention with China’s implementation of the rural revitalization strategy [61]. Research on the relationship of rural area development with resources and the environment has mainly been based on a single scale, at or above the county level, and seldom involved the agricultural areas in the oases of arid lands. Further, prior research lacked the comparison of scale differences and had less-pronounced correspondence relationships. Therefore, in this study, the coupled linkage relationships of planting industry transformation with pollutant discharge and water resource demand in oasis agricultural areas were established at the county and township levels, using various methods, such as inventory analysis, and coefficient and quota calculation. In addition, differences in resource and environmental pressures were compared under different spatial planting patterns. This helped verify the functional relationship of rural area development with resources and the environment at the micro scale, which can better reflect the characteristics of rural regional development. It also compensated for the lack of research on oasis agricultural areas, enriching the theoretical insights of rural geography. Ganzhou district represents the development of rural areas in oases in arid lands, and the sustainable development of its agriculture and rural areas is the epitome of oasis regions in China. The current research method and conclusion can propose transformation and development policies for similar rural areas, which contributes not only to the field but also to society.

The EKC provides an important theoretical basis for the relationship between regional development with resources and the environment, and has been empirically studied in most areas. However, not all areas have followed the U-shaped evolution path [62]. In some areas, the carrying capacity of resources and the environment could be improved with technological upgrading, economic investment, or industrial transformation when the carrying capacity reaches a threshold. This allows development to be re-coordinated with resources and the environment, exhibiting a continuously rising evolutionary relationship. Some areas have passively changed the development mode because of the limitations of resources and the environment, and presented an inverted “U”-shaped evolution path to adapt to environmental development. There have also been areas that took no measures, with a gradual decline occurring, owing to the limitations of resources and the environment. These showed an evolutionary process of continuous decline. Moreover, some areas exhibited coupled paths, such as “N” or inverted “N”, driven by policies [24,25]. The empirical study in Ganzhou district showed there was a complicated coupled linkage relationship of planting industry transformation in arid oases in rural areas with resource and environment pressures. This not only highlighted the scale differences and heterogeneity of spatial planting patterns but also showed a coupling evolution pattern for the coexistence of synergy and trade-off.

### 5.2. Sustainable Development Policy Recommendations for Oases

The oasis industrial structure urgently needs to be transformed and upgraded. Ganzhou district is the largest county-level corn seed production base in China. Corn seed production is a pillar industry of agricultural and rural economic development, as well as for increasing the income of farmers. In recent years, the benefits of the corn seed production industry have declined under the background of sustained economic downturn. This has made it increasingly difficult for farmers to continuously increase their income [63]. Local governments should adjust the planting structure, control the expansion of corn planting areas, tap the functions and values of villages driven by rural revitalization, cultivate a “planting+” diversified industrial system, and improve the livelihood resilience of farmers. Simultaneously, Ganzhou district is also a provincial-level key development zone, with an urbanization rate of 53.57%. It is a central city dominated by agriculture, with low-level functioning of population and economic gathering. It also has a weak industrial strength and a lack of job opportunities. These are the main reasons it cannot exercise its gathering function. In the future, we should focus on agricultural product processing to promote the integration of the three industries and urban–rural integration, improve the attraction of central cities, demarcate high-standard farmland red lines based on water resources, and transfer surplus labor force. This can reduce agricultural water consumption and the pollution pressure from the planting industry, while also reserving more water for ecology and other industries.

The contradiction between the supply and demand for water resources in the oases was prominent, and the exploitation and use of groundwater were high. The water level in Ganzhou district has dropped by 0.12 m annually, but the utilization efficiency of water resources is low. The lining rate of canal systems was only 0.6, and the output rate of water resources was only 70.3% in the inland river basin and 27.7% in the whole province. Controlling water consumption and improving the utilization rate of water resources are crucial for the sustainable development of agriculture and rural areas in oases [64]. Water and soil environmental pollution has been severe during the development of the oasis planting industry, whereas the proportion of sewage treatment and reuse was low, with Ganzhou district accounting for only 1.67% of the total water supply. Therefore, the construction of water-conservation and sewage treatment facilities should be strengthened, improving the level of agricultural modernization, reducing the dependence on agrochemistry, and paying special attention to the resource and environmental management of seed corn and vegetable planting areas at large spatial scales. In addition, the pollutants produced by crop straws gradually increased with the expansion of the planting area. It is, thus, necessary to educate and encourage farmers to take scientific measures to deal with straw. The evolution of resources and the environment is closely related to changes in national strategies, agricultural policies, and environmental regulations. It is important for local decision makers to consider the stages of agricultural production and rural area development, and formulate different laws and regulations according to the evolutionary track of resource and environment pressures. This can aid in regulating and restricting the behavior of people, improving resource use efficiency, and reducing the environmental pollution load.

### 5.3. Limitations of This Study and Outlook for the Future

This paper uses the officially released pollutant discharge coefficients and irrigation norms with regional characteristics to calculate the resource and environment pressures. However, the coefficient and norm will be affected by the progress of agricultural science and technology, and resource use efficiency. To date, there is no reliable literature that provides appropriate standards. Therefore, the results are complex and changeable, and there may be limitations. Due to spatial constraints and a lack of availability of long-term evolution data, this paper did not investigate the influencing factors on the relationship between the transformation of the planting industry and the resource and environmental pressures. An in-depth analysis of the driving mechanisms of this relationship is encouraged. These endeavors need subsequent empirical research after the collection of more detailed data.

## 6. Conclusions

Through studying the relationship between the planting industry transformation and the resource and environmental pressures, a scientific basis for policy guidance on the sustainable development of agriculture and rural areas in arid oases can be provided. Taking Ganzhou, an arid oasis agricultural region in China, as an example, this study confirmed the complex response characteristics of resource and environmental pressures with the change in planting scale and structure; that is, the expansion of crop planting scale increased the pressure on water resources demand, the average environmental pollution of cash crops was higher than that of grain crops, and different planting structures showed three environmental pollution trends, namely inverted U-shaped, continuous increase and continuous decrease. These results provide insight into the relationship between human activities and resources and the environment in arid oases, and provides a case study for rural environmental governance in similar areas in the world.

## Figures and Tables

**Figure 1 ijerph-19-05977-f001:**
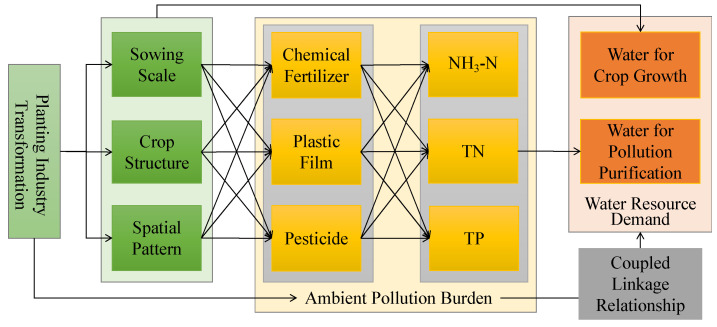
Evaluation framework of the relationship between planting industry transformation with resource and environment pressures.

**Figure 2 ijerph-19-05977-f002:**
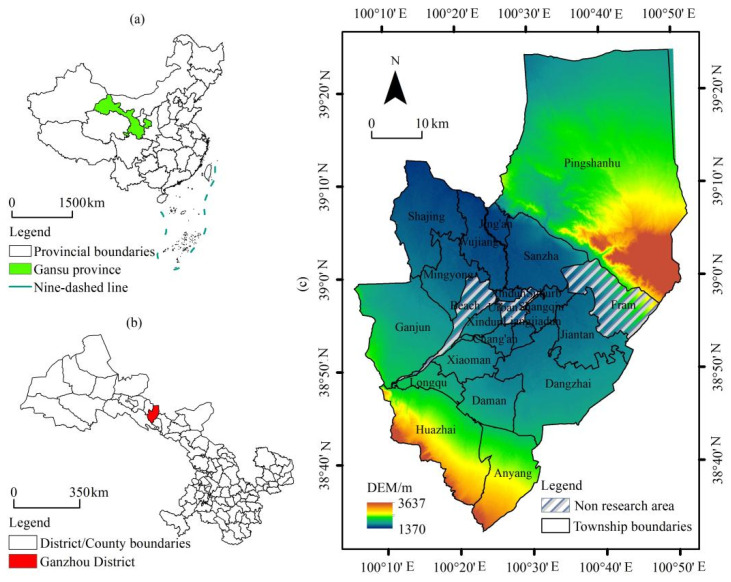
Location of (**a**) Gansu Province in China, (**b**) Ganzhou district in Gansu Province and (**c**) regional overview map of Ganzhou district.

**Figure 3 ijerph-19-05977-f003:**
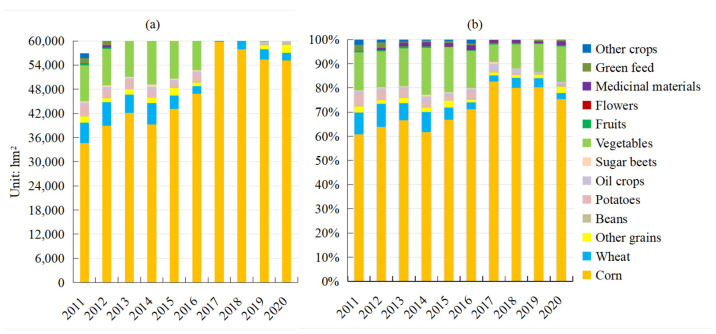
(**a**) Crop planting area and (**b**) proportion of grain crop change in Ganzhou district from 2011 to 2020.

**Figure 4 ijerph-19-05977-f004:**
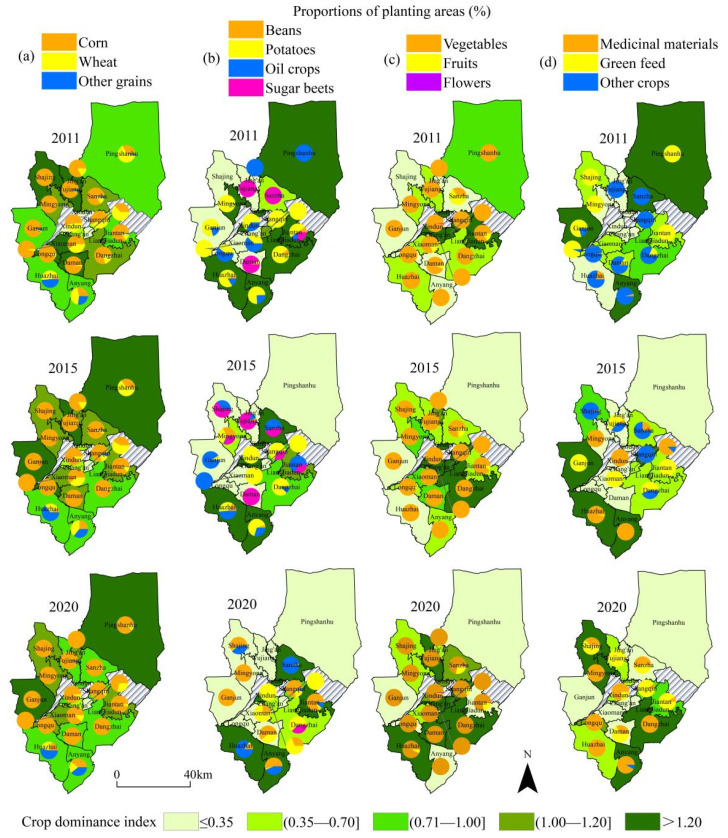
Spatial changes in the dominance of crop planting in townships from 2011 to 2020. (**a**) Proportions of corn, wheat and other grains; (**b**) Proportions of beans, potatoes, oil crops and sugar beets; (**c**) Proportions of vegetables, fruits and flowers; (**d**) Proportions of medicinal materials, green feed and other crops.

**Figure 5 ijerph-19-05977-f005:**
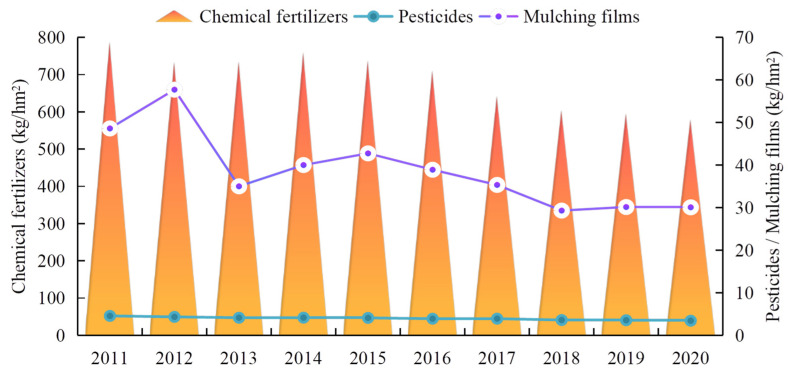
Consumption of chemical fertilizers, pesticides, and mulching films per unit area of cultivated land in Ganzhou district from 2011 to 2020.

**Figure 6 ijerph-19-05977-f006:**
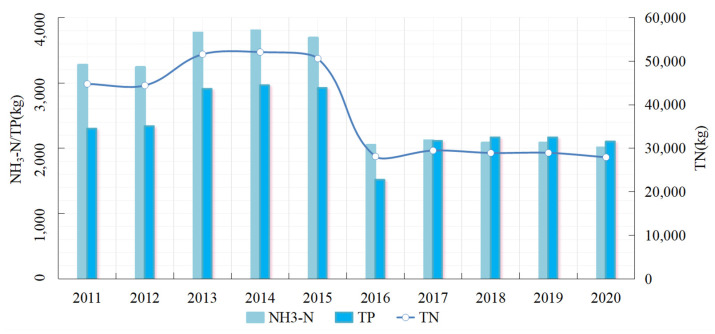
Changes in pollutant discharge loss from crop planting in Ganzhou district from 2011 to 2020.

**Figure 7 ijerph-19-05977-f007:**
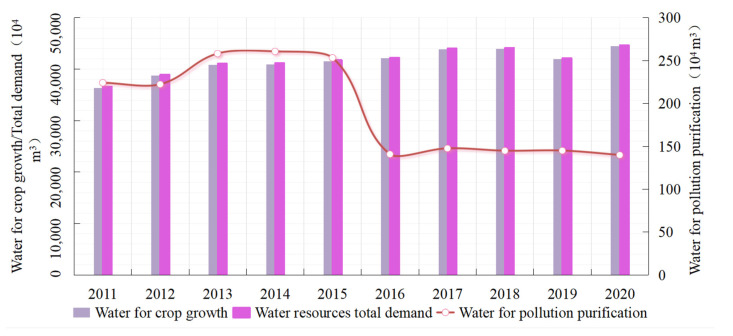
Changes in water demand for crop planting in Ganzhou district from 2011 to 2020.

**Figure 8 ijerph-19-05977-f008:**
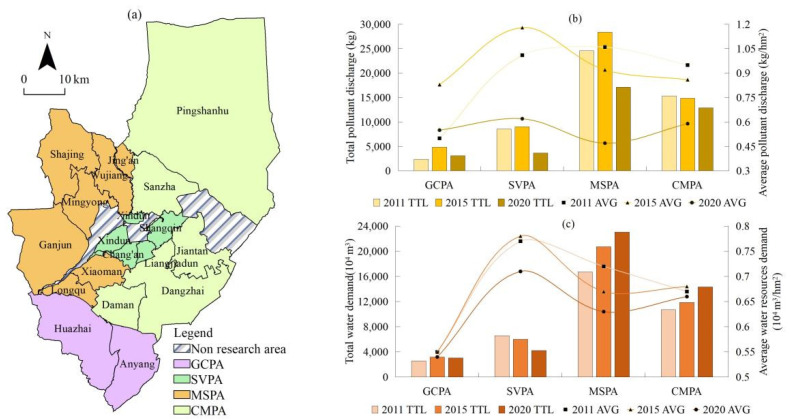
(**a**) Planting patterns of crops, (**b**) environmental pollutant discharge load and (**c**) water resources demand changes in Ganzhou district.

**Table 1 ijerph-19-05977-t001:** Nitrogen and phosphorus discharge loss coefficients from the planting industry in Gansu province.

Discharge Loss Coefficient of Crops during Sowing (kg/hm^2^)			Discharge Loss Coefficient in Garden Land (kg/hm^2^)		
Ammonia nitrogen	Total nitrogen	Total phosphorus	Ammonia nitrogen	Total nitrogen	Total phosphorus
0.029	0.403	0.029	0.039	0.493	0.018

**Table 2 ijerph-19-05977-t002:** Irrigation norm for crop growth in the Hexi area of Gansu province.

Crop	Irrigation Norm (m^3^/hm^2^)	Crop	Irrigation Norm (m^3^/hm^2^)
Rice	7500	Sugar beet	4725
Wheat	5250	Vegetables	6300
Corn	6150	Flowers and Plants	4800
Other grains	4800	Fruits	4800
Beans	5025	Medicinal materials	4725
Potatoes	3900	Fruit-bearing forests	3300
Oilseed	5025		

**Table 3 ijerph-19-05977-t003:** Relationship between transformation in the planting industry with resource and environment pressure responses.

Resource and Environment Changes	Response Intensity	Response Relationship	Coupled Linkage Pattern between Planting Industry Transformation and Environmental Pollution Lode/Water Resource Demand
∆RE > 0	RS > 0	Positive response	Synergetic evolutionary
	RS < 0	Negative response	Trade-off evolutionary
∆RE < 0	RS > 0	Positive response	Synergetic evolutionary
	RS < 0	Negative response	Trade-off evolutionary

**Table 4 ijerph-19-05977-t004:** Coupling characteristics of planting industry transformation with resource and environment pressures in Ganzhou district from 2011 to 2020.

Year	Planting Industry Transformation and Pollutant Discharge			Planting Industry Transformation and Water Resource Demand		
	∆Pollutant Discharge	RS Pollution Emission	Relationship	∆Water Resource Demand	RS Water Resource Demand	Relationship
2011–2012	−352.13	−0.10	trade-off	2350.93	0.91	Synergy
2012–2013	8268.66	4.39	Synergy	2128.11	1.45	Synergy
2013–2014	584.02	1.49	Synergy	104.20	0.38	Synergy
2014–2015	−1645.78	−1.86	trade-off	596.90	0.96	Synergy
2015–2016	−25,531.46	−22.93	trade-off	474.24	0.58	Synergy
2016–2017	2029.26	0.65	Synergy	1775.57	0.43	Synergy
2017–2018	−543.20	61.39	Synergy	99.70	−8.63	trade-off
2018–2019	36.51	−0.02	trade-off	−1983.78	0.93	Synergy
2019–2020	−1158.20	−0.55	trade-off	2501.29	0.94	Synergy
2011–2020	−18,312.33	−1.27	trade-off	8047.16	0.77	Synergy

**Table 5 ijerph-19-05977-t005:** Average pressure on resources and the environment under different crop planting patterns in Ganzhou district 2011–2020.

Planting Pattern	Included Townships	Crop Planting Characteristics
GCPA	Huazhai and Anyang	Townships with wheat, other grains, beans, potatoes, oil crops, medicinal materials, and other crops as dominant crops.
SVPA	Chang’an, Liangjiadun, Shangqin, and Xindun	Vegetables were the dominant crops, and the proportion of the planting area remained above 40%.
MSPA	Shajing, Ganjun, Wujiang, Mingyong, Xiaoman, Longqu, and Jing’an	Corn was the dominant crop, and the proportion of the planting area used for growing seed corn remained above 70%.
CMPA	Pingshanhu, Sanzha, Jiantan, Dangzhai, and Daman	While the planting area for corn continued to expand, the townships had other food and cash crops as the dominant crops.

## Data Availability

The data presented in this study are available on request from the corresponding author. The data are not publicly available due to restrictions eg privacy.

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
