# Peer review of "Resource and Environmental Pressures on the Transformation of Planting Industry in Arid Oasis"

_ijerph, 2022, doi:10.3390/ijerph19105977_

Round 1
Reviewer 1 Report
Title:
1/ condensate the main revelation into a short and groundbreaking claim
Abstract:
2/ better follow the established schema of writing academic Abstract: A/ introduction (urgency and significance of the research hypothesis); B/ principles of the methods used + key results; C/ conclusions (commercial and environmental impacts)
Introduction:
3/ review the phenomena from global perspective, do not limit the text to the case study
4/ make sure that this chapter fully introduces any reader into to the topic, explain all the terms, units, abbreviations and the whole context that is necessary for anyone (including experts from other disciplines) to understand the following chapters
5/ deeper review the latest trends in the field of planting, refer to paper "Techno-economic analysis reveals the untapped potential of wood biochar"
6/ the research hypothesis is not clearly stated, clearly justify the urgency and importance of its investigation, clearly identify those who will benefit from the findings
Materials and Methods:
7/ the method must be presented in such a way that it can be reproduced anytime, by anyone, anywhere (do not create obstacles like referring to specific location etc.)
Results:
8/ use only SI units if you want to get published in an international journal (avoid "mu" etc.)
9/ do not refer to any local names, this brings nothing to the global audience of readers
Discussion:
10/ show more self-criticism to your work (can all the methods and results be fully trusted? what are the weaknesses of the methods used? where do the main measurement inaccuracies arise? what are the limitations from a commercial point of view? are the lessons learned transferable to other fields?)
11/ avoid data overkill, present only the most most industrially important results
12/ propose some improvements and direction for future research, refer to paper "Silica Nanoparticles from Coir Pith Synthesized by Acidic Sol-Gel Method Improve Germination Economics"
13/ reveal the main driving mechanisms of your results, provide deeper synthesis and revel some more original/significant findings
Conclusions:
14/ do not repeat your methods and results again and again, please understand that the Conclusion chapter is not a summary of your work, present only original and industrially significant revelations that have the potential to expand the horizon of human knowledge (higher level of generalization is mandatory)
15/ clearly indicate whether the research hypotheses tends to be confirmed or not
Reviewer 2 Report
The manuscript presents original results contributing to understanding of the relationship between human and the environment in rural areas. It is generally well written. The results are clearly presented, quite well described and discussed and may be useful in practice.
The Material and Methods section should be written in more detail. The number of repetition should be given for each measurement.
Please describe the terms of performing the measurements in more detail.
Were the differences in the results presented in Table 4, Figures 5-6 statistically significant between individual years?
The authors wrote the correct conclusions.
Reviewer 3 Report
The authrors propose a study that analyses the industrial transformations of rural areas and their impacts on environment, adopting as case study a Chinese area. The study is interesting and the paper is of fair quality, with suitable methods and clear presentations of results. I have some suggestions to improve the paper, mainly related to the scientific presentation and structure. Please have a look to the commented MS in attachment.
